# The Future of Artificial Intelligence in Monitoring Animal Identification, Health, and Behaviour

**DOI:** 10.3390/ani12131711

**Published:** 2022-07-01

**Authors:** Jenna V. Congdon, Mina Hosseini, Ezekiel F. Gading, Mahdi Masousi, Maria Franke, Suzanne E. MacDonald

**Affiliations:** 1Department of Psychology, Faculty of Health, York University, Toronto, ON M3J 1P3, Canada; minah91@yorku.ca (M.H.); efgading@yorku.ca (E.F.G.); suzmac@yorku.ca (S.E.M.); 2Toronto Zoo Wildlife Conservancy, Toronto Zoo, Toronto, ON M1B 5K7, Canada; 3EAIGLE, Markham, ON L3R 9Z7, Canada; marsousi@eaigle.com; 4Toronto Zoo, Toronto, ON M1B 5K7, Canada; mfranke@torontozoo.ca

**Keywords:** artificial intelligence, orangutan, *Pongo abelii*, animal behaviour, animal health, identification, monitoring, conservation

## Abstract

**Simple Summary:**

Due to climate change and human interference, many species are now without habitats and on the brink of extinction. Zoos and other conservation spaces allow for non-human animal preservation and public education about endangered species and ecosystems. Monitoring the health and well-being of animals in care, while providing species-specific environments, is critical for zoo and conservation staff. In order to best provide such care, keepers and researchers need to gather as much information as possible about individual animals and species as a whole. This paper focuses on existing technology to monitor animals, providing a review on the history of technology, including recent technological advancements and current limitations. Subsequently, we provide a brief introduction to our proposed novel computer software: an artificial intelligence software capable of unobtrusively and non-invasively tracking individuals’ location, estimating position, and analyzing behaviour. This innovative technology is currently being trained with orangutans at the Toronto Zoo and will allow for mass data collection, permitting keepers and researchers to closely monitor individual animal welfare, learn about the variables impacting behaviour and provide additional enrichment or interventions accordingly.

**Abstract:**

With many advancements, technologies are now capable of recording non-human animals’ location, heart rate, and movement, often using a device that is physically attached to the monitored animals. However, to our knowledge, there is currently no technology that is able to do this unobtrusively and non-invasively. Here, we review the history of technology for use with animals, recent technological advancements, current limitations, and a brief introduction to our proposed novel software. Canadian tech mogul EAIGLE Inc. has developed an artificial intelligence (AI) software solution capable of determining where people and assets are within public places or attractions for operational intelligence, security, and health and safety applications. The solution also monitors individual temperatures to reduce the potential spread of COVID-19. This technology has been adapted for use at the Toronto Zoo, initiated with a focus on Sumatran orangutans (*Pongo abelii*) given the close physical similarity between orangutans and humans as great ape species. This technology will be capable of mass data collection, individual identification, pose estimation, behaviour monitoring and tracking orangutans’ locations, in real time on a 24/7 basis, benefitting both zookeepers and researchers looking to review this information.

## 1. Introduction

### 1.1. The History of Technology

#### 1.1.1. Chronological Background

Over time, technology for monitoring both human and non-human lives has evolved and advanced. Dating back to the 1970s, collars with electronic transponders were attached to cows to automatically record data [1,2]; since then, technology has become smaller and more affordable, with a focus on improving overall productivity in agriculture [3,4]. These types of devices are now less intrusive and commonplace in the home, allowing humans to monitor their pet companions (e.g., smart home cameras) [5]. Although contemporary technology has a wide range of uses, the focus of this paper is on zoo management to enhance non-human animal welfare. We provide a brief overview regarding the types of hard- and software currently available and implemented in both the wild and captivity, and address remaining gaps in zoo-focused technology.

Overall, technology has been critical to obtaining large amounts of data, greatly reducing the labour-intensive and invasive aspects of data collection. Not only has technology reduced time in conducting research, but it has allowed for more accurate collection compared to that of human observers. For example, when Desormeaux and colleagues [6] compared the information collected from motion triggered cameras located at migratory fence gaps to observers’ recordings of mammal tracks (e.g., giraffe, zebra, elephant, and hyaena), a higher volume of crossing was reported with technology, suggesting that human observers may often miss important events. Methodologies that use manual collection of data are often time consuming, time limited, and potentially invasive and/or ill-suited depending on the research focus, supporting further investment in the advancement and refinement of technology-based alternatives.

#### 1.1.2. Locations of Employed Technology

In addition to footprints used to collect an estimate of the number of animals in a particular geographical location, to further gather information from wild animals researchers have collected hair, urine, fecal matter, etc. (e.g., [7,8,9]). Although minimally invasive or noninvasive, these types of data collection can be tedious and limited, requiring consideration of both accuracy and potential confounding variables in interpreting results. As a result, a wide range of technology has been employed, often in wild settings. For example, accelerometers have been extensively employed because they are inexpensive, animal-attached tags used to remotely gain positional data and 3D reconstructions of behaviour of often “unwatchable” animals [10,11]. Visibility issues can also be combated with another form of technology: drones. The use of remote-controlled or software-controlled flying robots have revolutionized monitoring of many marine species, including sharks [12], rays [13], dolphins [14], and seabirds [15]. Additional forms of technology include radiotracking and GPS collars, cameras, and artificial intelligence, used to detect lions (*Panthera leo*) [16], lynx (*Lynx lynx*) [17], giant pandas (*Ailuropoda melanoleuca*) [18], wild red fox (*Vulpes vulpes*) [19], and a range of primate species including blond capuchins (*Sapajus flavius*) [20], baboons (*Papio ursinus*) [11], and chimpanzees (*Pan troglodytes*) [21]. These projects have tracked physical locations of animals, with some capable of gathering information from individual animals, including behavioural data; i.e., face detection to record locomotor behaviour [16], camera traps to capture a range of daily activities [19], and tracking collars to quantify a range of daily activities [11].

However, there are many situations in which animals live in captivity under human care. As mentioned, when considering the development of technologies across the globe, many of the advancements have been produced for agricultural applications, which have clear economic advantages for humans. In captive settings outside of agriculture, monitoring vulnerable individuals (e.g., endangered or injured) is not only critical to ensure the health and well-being of such animals, but it can provide information to inform greater animal management decisions. The type of data researchers collect from wild animals is quite different from data collected from animals in captivity as the locations of captive animals are limited within relatively unchanging and predictable environments, and the individuals are known and remain in place across months, years, or decades. Thus, captive animal data are typically more welfare related, useful in informing animal husbandry and management decisions. Accredited zoos, aquariums, and conservation areas employ the best practices to monitor their animals and obtain necessary behavioural data however possible. For example, the low-cost “ZooMonitor” application has allowed for the tracking of pygmy hippos (*Choeropsis liberiensis*) and domestic chickens (*Gallus gallus*), providing data on physical appearance, habitat use, and behaviour within their enclosures [22]. Formerly expensive, intrusive, and inaccurate technologies have also recently been replaced by software and complex algorithms; for example, deep neural networks are computer technologies that have “learned” to detect and recognize individual giant pandas [23]. (Note: These neural networks are described in more detail in Section 1.2.2)

### 1.2. Recent Technological Advancements

#### 1.2.1. Purposes for Technological Advances

Across the many countries that technology has been developed in order to monitor animals, the variety of purposes for implementation include monitoring health (e.g., [24,25]), differentiating between individual animals (e.g., [18,26]), and behaviour (e.g., [11,19,27]). A focus on health is the primary rationale for much of the technological advances in agriculture, given the focus on economic outcomes. Variables such as head motion, core body temperature, and heart rate [28], as well as surrounding temperature and humidity [24] are crucial for agricultural production. For example, in cattle, stress due to warm weather can influence vulnerability to disease, food consumption, weight gain, reproduction, and milk production [24,25]. Thus, in being able to track individual animals, technology allows for the monitoring of changes in livestock [3]. Monitoring individual animals is also integral to generating inferences about spatial ecology. For example, GPS tracking collars have been critical to mapping the urban movement patterns of raccoons [29], while camera traps have successfully tracked individual pandas amongst dense bamboo, eventually to be used to estimate population sizes [18]. Automated facial recognition has also been developed to identify individual wild gorillas, useful for evaluating spatial biodiversity in the wild [26]. In addition to monitoring health and differentiating between individuals, technology has advanced to be able record the actions of animals.

Only recently has technology been able to monitor animal behaviour as it is a complicated process; not only does the technology have to be able to recognize the individual from their environment, but must be capable of interpreting similar patterns of movements as distinct behaviours (e.g., agonistic behaviour vs. play). Traditional behaviour research involves many hours of training researchers and volunteers to correctly identify animal behaviour based on clear, detailed ethograms. Similarly it takes many hours and thousands of images to train artificial intelligence to “recognize” distinct behaviours. The payoff of this technology is that it allows for longer durations of monitoring, without the need for human observers, resulting in mass data collection [30]. Artificial intelligence can also be applied to large groups of animals (e.g., ant colonies and bee hives [31]), or implemented for monitoring animals that are often difficult to sample, namely cryptic species such as the nocturnal spotted-tailed quoll [32] or species that cover an extensive range such as seabirds [27]. Historically, collecting information from these species has proven difficult, but technology allows for the prediction of breeding biology and at-sea foraging behaviour, respectively [27,32]. In summary, much of the information that we learn from species would not be possible without these technological advancements.

#### 1.2.2. Technology Types

Evidently, a variety of technologies are available, such as wearable sensors, described above for livestock, that can track individuals and monitor changes [33,34]. However, it is important to note that, as the title suggests, these devices are quite obtrusive and sometimes invasive. Research on wild animals instead often involves the deployment of camera traps, which are triggered by wild animals passing by [18,20,35], or using thermal-sensing technology [32,35,36]. In the past two decades, facial recognition technology has developed, allowing for unobtrusive individual recognition and tracking [16,21,37,38]. With recent advancements in technology, computer software involving machine and deep learning, as well as artificial neural networks (ANNs), have been trained to provide surveillance of animals with regard to their location, species identity, individual identity, and/or behaviour (e.g., [19,21,27,30,31,39,40,41,42,43,44]). Machine learning focuses on algorithms and data to imitate the way humans learn; deep learning is a subfield of machine learning. Machine learning involves training computers to function to perform a task with minimal human intervention (i.e., explicit programming), whereas deep learning involves training computers to “think” using a human-like brain structure. The process and output of deep learning is more complex and requires larger datasets for training. ANNs mimic human brains through a set of algorithms. These ANNs consist of “layers” and require more than two layers to be classified as deep neural networks (DNNs) [45].

### 1.3. Current Limitations and Proposed Novel Technology

As outlined above, many technologies have been designed and developed, currently capable of recording animals’ identity, movement, heart rate, and temperature. Unfortunately, wearable devices can be intrusive and invasive when physically attached to the monitored animal, and evidence suggests that their presence may affect individuals’ behaviour (e.g., transmitters on snakes [46]; radio tags on birds [47]). Further, there are many species for which such devices are not compatible (e.g., some animals are too small or too large—GPS collars cannot be fitted on male polar bears because their necks are larger than their heads [48]), whereas camera technology is not capable of collecting all of the same information remotely (i.e., one technology capable of movement, identity, and temperature). While these technologies all provide critical advancements in monitoring animals, upon review of the literature at the initiation of this project, there appeared to be no single technology capable of providing all of the above information, both unobtrusively and non-invasively. Canadian tech mogul EAIGLE Inc. (hereafter “EAIGLE”) has developed and produced artificial intelligence (AI) software solution capable of determining where people and assets customers are within industry partners’ stores or public places or attractions for operational intelligence, security and health and safety applications. The solution also improves layout and monitors individual temperatures to reduce the potential spread of COVID-19 (Figure 1). Conservation of highly endangered species is a critically important issue in these days of climate and anthropogenic change. Modern zoos, such as the Toronto Zoo, are at the forefront of efforts to save species from extinction, and to ensure that animals in their care live enriched, healthy lives in species-typical environments. It is evident that mass collection of all of multiple sources of data points would provide zookeepers, conservation workers, and researchers with the necessary information to learn more about the species, and individuals under their supervision, in order to provide the best care attainable while exploring further reintroduction and conservation possibilities. Presently, research on animal behaviour is typically collected by observers collecting real-time data via paper, pen, and clipboard, or videos that are later manually annotated (i.e., labeled). Unfortunately, this type of data collection can be a lengthy and tedious process, and observations are limited to the hours in which observers are present.

With consideration of EAIGLE’s AI software, a novel opportunity arose for an industry partner (EAIGLE), an academic institution (York University), and a not-for-profit institution (Toronto Zoo Wildlife Conservancy) to become partners in developing an AI that fills the gap in current technology: tracking orangutans’ location in real time, identifying individuals, pose estimation (i.e., specifying body landmarks, e.g., shoulder and head), and behaviour monitoring to a high level of specificity. Given that the AI was originally developed for use with humans, we initiated this project with a focus on our close relative, the critically endangered Sumatran orangutan (*Pongo abelii*). The research team members are collecting, preparing, and analyzing the images from the orangutans in their zoo enclosure, and EAIGLE is training the AI model. The purpose of the AI model is to identify individuals and to analyze their behaviours, as well as their body temperature, and limb/joint position.

Orangutans are members of the same great ape taxon as humans, diverging from us in the evolutionary tree only 12–16 million years ago (e.g., [49,50]). Studying orangutans can provide a strong comparative model for the evolution of complex cognitive abilities, including perception, memory, and sociality. Orangutans have a unique ecological and behavioural repertoire, resulting in some important differences between these two species. Orangutans are: (1) non-vocal learners (i.e., do not learn species-specific vocalizations from a parent or teacher), (2) arboreal (i.e., tree dwelling), and (3) semi-solitary (i.e., social bonds are more easily formed between adult females and weaned offspring). Sumatran orangutans are critically endangered (~7,500 remain) [51] and the future of this species consists of living in zoos and conservation sites. Because these animals spend their lives in human care, it is important to understand how the animals interact with their artificial environments, including where they spend their time and which behaviours they complete within a day. For example, where are animals active vs. resting within their habitat? Are there changes in foraging, urination, and defecation indicating a health issue? Are there any increases in agitated, agonistic, or stereotypical behaviours indicating an increase in stress? Understanding how animals interact with their environment is useful for making changes in the layout of their habitat, ensuring a species-typical lifestyle, and improving both group and individual enrichment. Changes in behaviour from baseline, however, can indicate potential health or welfare issues with an individual or provide general data that could assist with husbandry improvements. Therefore, this artificial intelligence will greatly improve animal welfare outcomes. The ability for location detection and individual recognition, for instance, will be critical to monitoring pregnant or sick animals, and track development from infancy to old age. An ability that many technologies lack is pose estimation; the capacity to track body position is not only necessary to flag any potential limb issues, but behaviours are defined by distinct patterns of movements determined by the position of the limbs and joints. All of these capacities combined will ensure for artificial intelligence that addresses the current gap in available technology.

## 2. Materials and Methods

### 2.1. Dataset

For Phase 1 of this project, a dataset of 20,000 images was collected in which 5000 images were collected from online resources and 15,000 images were collected from fixed cameras that are installed inside the orangutan habitat at the Toronto Zoo to ensure that the trained model would be able to detect orangutans (see Figure 2 for examples). Five 12 MP cameras with variable focus lenses were installed inside the orangutans’ habitat in order to provide nearly complete coverage of the habitat, minimizing blind spots (see Figure 3 and Figure 4 for panoramic photo and blueprint of the enclosure and camera locations, respectively). These stationary cameras are capable of collecting image stills (Phases 1 and 2) and video clips (Phase 3). Images included multiple orangutans in a single scene which allowed for training of the AI in order to recognize the potential for multiple orangutans visible on camera.

Since there was no publicly available pre-trained deep learning model for orangutan detection, we used generic object detection and recognition deep neural network models, including YOLOv4, EfficientNet, and Efficient Pose, which are used for orangutan/zookeeper/visitor detection, orangutan recognition, and behavioural analysis based on pose alignments and movements [52,53,54]. However, to train these models, we first needed to collect a dataset. Therefore, we started with using a method for object detection based on change detection in the background image. A change in the scene corresponds to an object movement. Any changes in the background image with a minimum of 50x50 connected pixels area in the image was taken as a non-noise change in the image; images were taken from each of the five cameras at the frequency of 30 frames per seconds. Once a change was detected, the images were selected and stored in a 2 TB hard drive. The software ensured that at least a 30 s delay existed between two consecutive frames, taken from each camera. Once the images were collected, we refined them by removing redundant images or images without objects of interest. This measure was taken to ensure that collected images were distinct enough from each other to avoid any bias in the dataset.

Two annotation programs were developed: one of the annotation programs was designed to draw bounding boxes for each subject (i.e., orangutan, zookeeper, or zoo visitors), whereas the second one was used to segment each part of the orangutans’ body and specify body landmarks (i.e., header, neck, shoulder, elbow, hand, hip, knee, etc.). In Phase 2, 15,000 images were annotated and the specific names of each of the six (6) Toronto Zoo orangutans (4 female/2 male, range 15–54 years old) were recorded within every image’s annotation (EfficientNet model used for Phase 2 recognition; see Table 1 for orangutan names and defining characteristics used to identify each individual; see Figure 5 for an annotated example image during training). Thus, following Phase 1′s image collection and orangutan detection (i.e., distinct from their habitat), the technology was further trained to recognize individual orangutans, distinct from zookeepers, visitors, and other orangutans. Phase 2 was essential as individual identification will be critical to provide personally adapted care for each individual based on the information collected; see Figure 6 for a screen capture of the current technology, successfully identifying the location of individual ‘Puppe’.

### 2.2. Implementation

Each phase further trains the AI model to be able to recognize orangutans’ location, identity, physical stature, and all executed behaviours. Presently, we are in the process of completing Phase 3 (see Figure 7 for a flowchart of the project phases). This third phase includes pose estimation and annotating the behaviours of orangutans including foraging, brachiation, locomotion, object play, object manipulation, fiddling, scanning, patrolling, hiding, inactivity, urination, defecation, agitated movement, affiliative vs. agonistic behaviours, and keeper-, guest-, self-, baby-, and tech-directed behaviours (see Table A1 for a full ethogram of orangutan behaviours; note: baby-directed behaviours have recently been added due to the April 2022 birth of an infant male, mothered by Sekali and fathered by Budi). This level of data collection will be timely, including annotating series of brief video clips and specifying body landmarks using our second annotation tool. Due to the importance and time-consuming nature of this process, this phase will be the most critical piece of this project.

The developed algorithm is capable of merging repetitive objects in the event of overlap between cameras. The developed algorithm is able to identify the same object detected by multiple cameras. This is performed by a similarity check of the objects detected by multiple cameras. The similarity check is performed using both the location of the object and features extracted using deep convolutional network from detected object. It provides a feature vector including 64 values of the object which compresses the appearance information of the animal. Using the similarity check, the algorithm merges multiple detections of each single animal in the habitat.

## 3. Results and Discussion

At this time, EAIGLE technology is still in the process of being trained to the highest accuracy. The AI is already successful in detecting orangutans from the background, distinguishing between orangutans vs. zookeepers vs. zoo visitors, and correctly labeling each of the six orangutans (refer to Table 1). Current rates of accuracy have been assessed and are high; specifically, the accuracy of detection is 94% when the animal is at a minimum size of 50 × 50 pixels, whereas the accuracy of recognition on an individual basis is 80% per single image and increases to 92% with majority voting over 50 frames.

Currently, the AI is being trained to monitor pose estimation and classify animal behaviour (refer to Table A1 for ethogram), a critical and lengthy phase of calibrating this technology. This product will be capable of mass data collection, which will not only benefit the animals at the Toronto Zoo, but which will, we hope, become the ‘gold standard’ for zoos worldwide.

Our proposed AI addresses a current gap in technology for unobtrusive, 24/7 monitoring non-human animals in collecting a wide-range of information. In understanding where animals spend their time, zoo staff can rearrange their enclosures to be more suitable, comfortable, and cognitively stimulating, improving enrichment and decreasing any detrimental impacts on these captive animals (e.g., [55]). In large conservation areas, gamekeepers could monitor the location and health of the animals (e.g., alarmingly high or low temperatures). Therefore, this computer technology will ensure accurate, unbiased, reliable, and rapid data collection to address a multitude of research questions, from applied welfare to comparative evolution and cognition. Data from captive-based studies such as these will also be useful to provide baselines for species that have yet to be studied extensively in the wild.

## 4. Conclusions

When the AI is fully validated, we propose extending this project to record and analyze the behaviours of other non-human animals, including (quadrupedal and mammalian) tigers and polar bears, which are also critically endangered. We expect that this technology will be adaptable upon extension to other species. This can be approached in two (2) ways: (1) training a separate model for each animal, or (2) training the current model to detect animals and their species. Considering the variation of characteristics between animals, we recommend grouping animals which are usually placed in the same enclosure and training a model based on the data collected to reduce the complexity of training deep learning models. This helps to keep the size of deep learning models small which allows them to run on small on-edge processors. Additionally, this helps to increase the accuracy of classifying species in a habitat. In addition, both of these species (tigers and polar bears) are also typically solitary, making the identification process easier. This technology is capable of locating orangutans within their indoor habitat (The Indomalayan Pavilion), distinct from foliage and background movements (e.g., birds, visitors, and keepers). In addition, this AI can be accommodated for the tigers and polar bears to distinguish them from their surroundings. Lastly, orangutan behaviour is considered to be quite complex due to the dexterity of their limbs; tiger and polar bear behaviour could be classified as more simplistic, but they make many fine motor movements that may be difficult to capture on camera. Eventually, this AI will be applicable for large mammals throughout the Toronto Zoo, can be implemented in other major zoos in North America and beyond, and eventually modified to implement in conservation areas for monitoring endangered wildlife.

## Figures and Tables

**Figure 1 animals-12-01711-f001:**
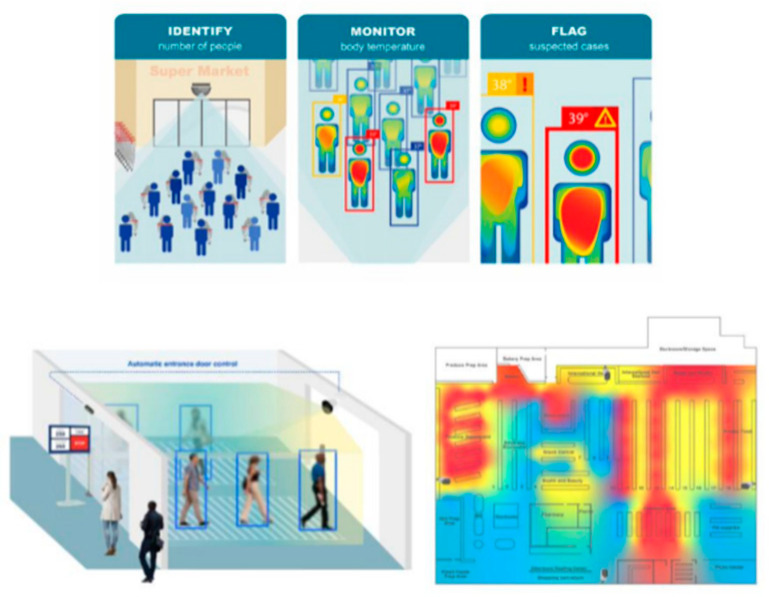
An image of EAIGLE’s current artificial intelligence that can identify the number of people in a room (top left), monitor individual body temperature (top center), and flag suspected cases (top right) of COVID-19. This allows the technology to monitor crowds of people and count them (bottom left), while mapping their real-time density on the facility layout (bottom right) as deployed in high-foot traffic facilities and public places.

**Figure 2 animals-12-01711-f002:**
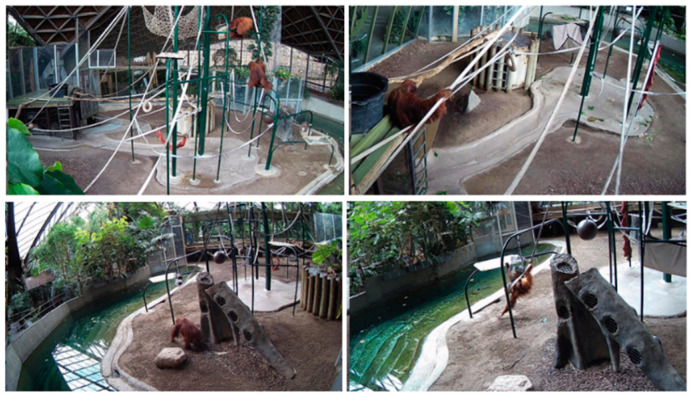
Examples of images collected from five cameras installed in the orangutans’ habitat in Toronto Zoo.

**Figure 3 animals-12-01711-f003:**
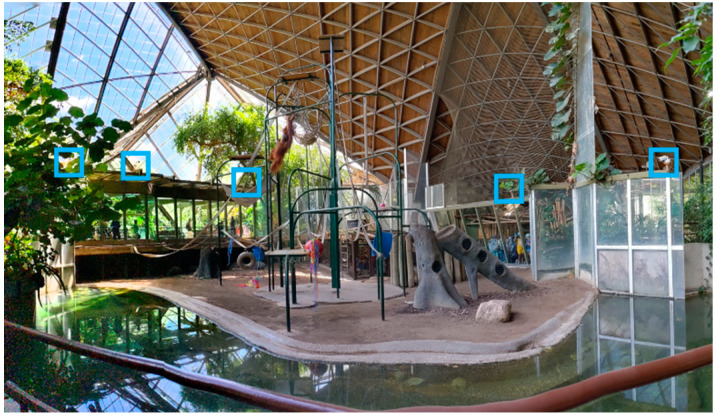
A panoramic photo (Google Pixel 5) taken of the indoor orangutan exhibit. Blue boxes indicate the location of each of the five (5) cameras around the perimeter of the enclosure. Orangutan (Sekali) can be seen brachiating between bars.

**Figure 4 animals-12-01711-f004:**
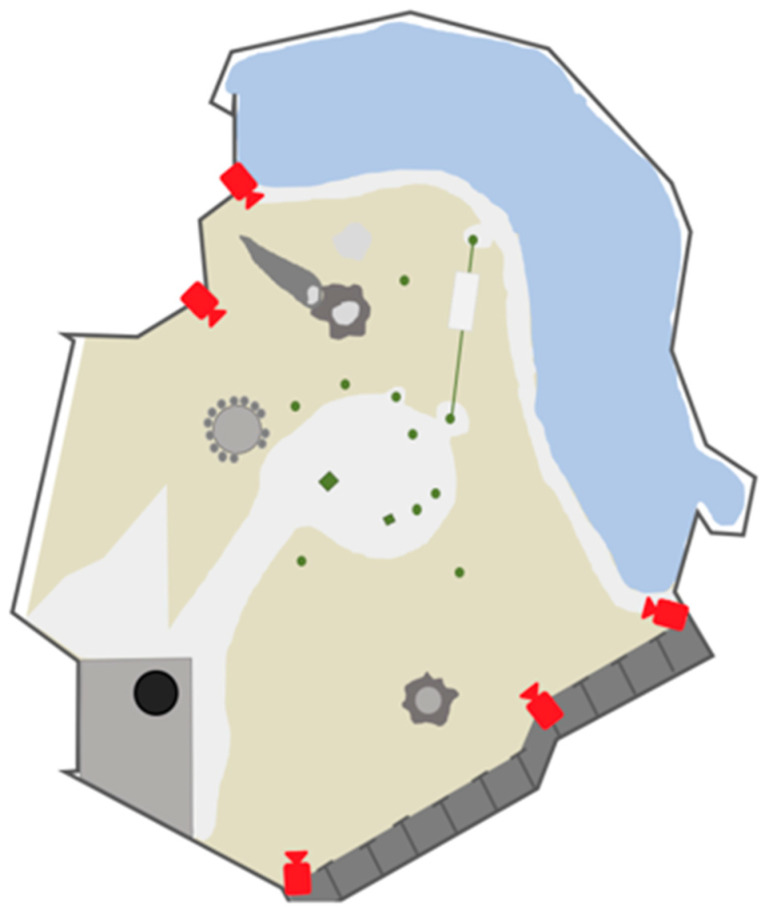
A blueprint of the indoor orangutan exhibit study area. The location of each of the five (5) cameras is indicated above in red. Beige (middle) indicates the dirt ground, light grey (center) indicates the cement ground, dark grey (bottom) indicates to the raised walkway and platform, and blue (upper right) indicates the water of the moat. Additional grey shapes (throughout) signifies the location of enrichment objects, with green dots referring to the base of the jungle gym bars. Note for scale: Each section of the dark grey walkway is 6 ft wide.

**Figure 5 animals-12-01711-f005:**
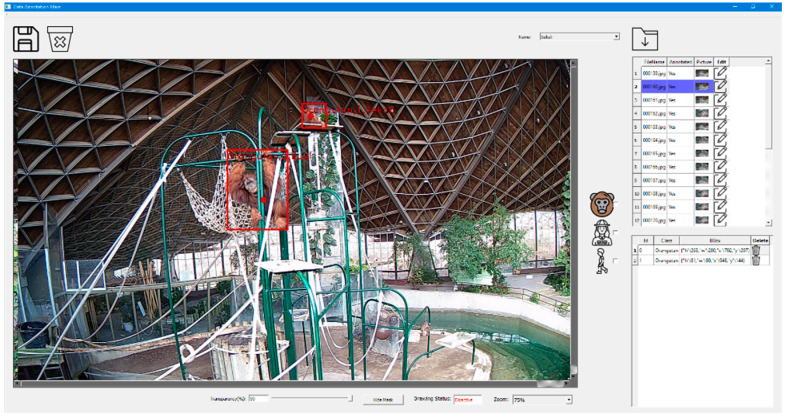
An example of an annotated image for training the artificial intelligence model to identify Sumatran orangutans (“Budi”, lower; “Sekali”, higher). The displayed tool allows for a choice between annotating orangutan vs. zookeeper vs. guest, and identification between the six orangutans.

**Figure 6 animals-12-01711-f006:**
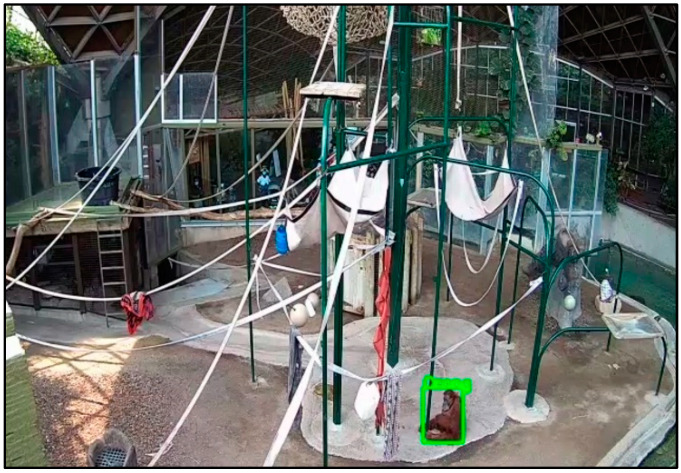
Screen capture of the artificial intelligence detecting an orangutan (“Puppe”). Characteristics include facial details (eyes, mouth, colouration), hair colour and consistency, etc., as compiled from thousands of images. Refer to Table 1 for a list of distinct, individual characteristics that raters used to annotate these images.

**Figure 7 animals-12-01711-f007:**
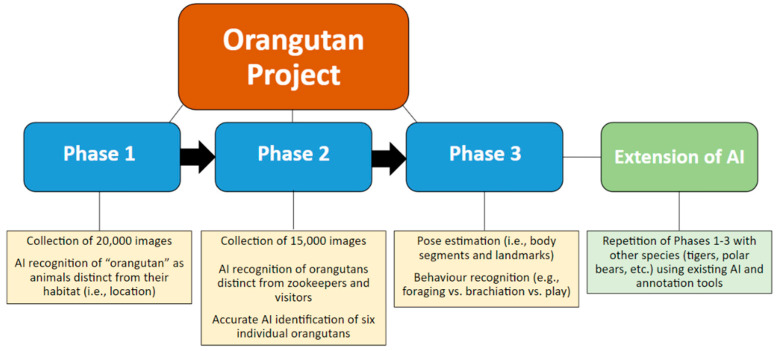
A flowchart of the orangutan project and extension of the resulting artificial intelligence (AI). The details of each of the three (3) phases of the orangutan project are provided, along with a brief explanation of the extension.

**Table 1 animals-12-01711-t001:** Toronto Zoo Sumatran orangutan information, including name, birth date, current age (range 15–54), biological sex (F = female; M = male), birth place, and defining features.

Orangutan Name	Birth Date	Current Age (June 2022)	Biological Sex	Birth Place	Defining Features
Puppe	1967/09/07	54	F	Sumatra	Dark face with dimpled cheeks; yellow hair around ears and on chin; reddish-orange hair, matted hair on back/legs; medium body size; always haunched with elderly gait/shuffle; curled/stumpy feet
Ramai	1985/10/04	36	F	Toronto Zoo	Oval-shaped face with flesh colour on eyelids (almost white); red hair that falls on forehead; pronounced nipples; medium body size
Sekali	1992/08/18	29	F	Toronto Zoo	Uniformly dark face (cheeks/mouth), with flesh coloured upper eyelids and dots on upper lip; horizontal lines/wrinkles under eyes; long, hanging, smooth orange hair with “bowl-cut”, and lighter orange hair to the sides of the mouth; mixed light and dark orange hair on back (light spot at neck); medium-sized body; brachiates throughout the enclosure
Budi	2006/01/18	16	M	Toronto Zoo	Wide, dark face with pronounced flanges (i.e., cheek pads); dark brown, thick/shaggy hair on body and arms, wavy hair on front of shoulders; large/thick body; stance with rolled shoulders; often climbing throughout enclosure
Kembali	2006/07/24	15	M	Toronto Zoo	Oval/long face with flesh colour around eyes and below nose; dark skin on nose and forehead; small flange bulges; hanging throat; reddish-orange shaggy hair with skin breaks on shoulders and inner arm joints; thicker hair falls down cheeks; large/lanky body; strong, upright gait; often brachiating throughout the enclosure
Jingga	2006/12/15	15	F	Toronto Zoo	Oval/long face with flesh colour around eyes and mouth; full lips; reddish-orange hair with skin breaks on shoulders, inner arm joints, and buttocks; thicker hair falls on forehead and cheeks; medium-small body size

## Data Availability

Not appliable.

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
