# Peer review of "The Future of Artificial Intelligence in Monitoring Animal Identification, Health, and Behaviour"

_animals, 2022, doi:10.3390/ani12131711_

Round 1
Reviewer 1 Report
The authors of this study are dedicating their efforts toward a very promising field regarding captive care. I was very positively surprised to which extend they managed and plan to further improve their system and agree with the authors regarding the enormous potential for care and welfare purposes as well as research opportunities. Thus, the project itself and the presented study is definitely worth being published in this journal.
I only recommend here some minor modifications to make this paper easier to understand as well as even more attractive to readers and potentially interested people who might want to make use of said AI-camera system.
Introduction section:
102 “Captive animals are almost always visible” – I suggest to rephrase this. In comparison to wild animals they are definitely more visible but there are still plenty of captive situation were visibility is an issue for observations due to off-display areas and/or complex and lush outdoor enclosures. However, this could be easily solved by the suggested camera system.
230 Just a minor detail: I was surprised to see you comment “play vs. resting”. I don´t see why those two should be compared, it would seem more logical to so “activity vs. resting” for example.
Materials and method section
In the material and method section I do miss information regarding the observed population (which strangely is mentioned in the table in the result section) and especially the enclosure dimensions. This should be added at least briefly to better understand how 5 cameras are able cover said enclosure and how many cameras would be need for bigger or more complex installations. It might also be useful for the reader to see a blue print of the enclosure and how the cameras cover the areas marked on said blue print.
Also not totally clear on if the cameras have an overlap of their covered areas or if cameras have to be installed in a way that animals cannot simultaneously appear in frames collected by two different cameras at the same moment.
Please clarify if the 20.000 frame were necessary purely to train the AI to identify individuals and surrounding parameters or was used for other purposes as well already?
You explain in line 248 about phase 1 and in line 293 about phase 3, however I do not find information regarding phase 2 – this needs to be clarified.
Result section:
I don´t understand why “Table 1” is in the result section. This should be moved to Material Section where it is mentioned for the first time within the text or should be placed as a supplementary table.
I suggest to add more information about its future usefulness in populations that are not housed at the Toronto Zoo. For example, I am not clear on how much of the AI training was dedicated to the “individual and surrounding of the enclosure” identification and how much was dedicated to improve the accuracy of general orangutan, posture and behavior identification. By adding this information many readers might get a clearer Idea and decide on if this technology is actually something, they might be interested in installing in other populations.
I also suggest to add a final paragraph explaining the potential and hurdles that need to be overcome before it can be used in other species, although I understand you are still working on phase 3 for orangutans.
Reviewer 2 Report
Although information was given about data collection in phase 1 in the study, it did not provide any information about the procedures performed in phase 2. In particular, information such as what the selected AI method is and which parameters are used in the "Implementation" section are missing. In order for the study to be better understood by the readers, it would be appropriate to give a flow diagram showing which phases the study consists of and what has been done or will be done in these phases.
